# Expert consensus on moving towards a value-based healthcare system in the Netherlands: a Delphi study

Gijs Steinmann ![ORCID],[1] Diana Delnoij,[1] Hester van de Bovenkamp,[1] Rogier Groote,[2] Kees Ahaus[3]

[1]Health Care Governance, Erasmus Universiteit Rotterdam Erasmus School of Health Policy and Management, Rotterdam, Zuid-Holland, Netherlands
[2]Department of Operations, Faculty of Economics and Business, University of Groningen, Groningen, Netherlands
[3]Erasmus School of Health Policy & Management, Department Health Services Management & Organization, Erasmus University Rotterdam, Rotterdam, Zuid-Holland, Netherlands

**Correspondence to**
Mr Gijs Steinmann;
steinmann@eshpm.eur.nl

## ABSTRACT

**Objectives** While the uptake of value-based health care (VBHC) is remarkable, uncertainty prevails regarding the most important actions and practices in establishing a value-based healthcare system. In this paper, we generate expert consensus on the most important aspects of VBHC.

**Design** The Delphi technique was used to reach consensus on the most important practices in moving towards a value-based healthcare system.

**Setting and participants** A Dutch expert panel consisting of nine members participated in a two-round survey.

**Primary and secondary outcome measures** We developed 39 initial items based on the pioneering literature on VBHC and recent health policies in the Netherlands. Experts rated the importance of each item on a 4-point Likert scale. Experts could change items or add new ones as they saw fit. We retained items that were rated (very) important by ≥80% of the panel.

**Results** After two survey rounds, 32 items (72%) were included through expert consensus. Experts unanimously agree on the importance of shared decision-making, with this item uniquely obtaining the maximum score. Experts also reached consensus on the importance of outcome measurements, a focus on medical conditions, and full cycles of care. No consensus was reached on the importance of benchmarking.

**Conclusion** This paper provides new insight into the most important actions and practices for establishing a value-based healthcare system in the Netherlands. Interestingly, several of our findings contrast with the pioneering literature on VBHC. This raises the question whether VBHC's widespread international uptake indicates its actual implementation, or rather that the original concept primarily serves as an inspiring idea.

## INTRODUCTION

Value-based health care (VBHC) is a highly topical concept within many healthcare systems.[1–3] The concept was pioneered by Michael Porter and Elizabeth Teisberg, who propose an overarching goal for all stakeholders in health care: to optimise value for patients.[4] Thus far, however, it remains relatively unclear how to transition this popular idea into the actual establishment of a

### Strengths and limitations of this study

► Using the Delphi technique, this study generates expert consensus on the most important actions and practices in moving towards a value-based healthcare system.
► By revealing 32 actions and practices, this research operationalises value-based health care, a highly abstract and multifaceted concept.
► Although the selection of experts was appropriate for the purpose of this study, the results may have limited generalisability.
► The importance attached to specific aspects of value-based health care may be subject to change, with some attaining relatively more or less importance depending on a particular timeframe.
► Experts could reformulate existing items and also suggest new ones; this enabled participants to express their personal (re)interpretation of value-based health care.

value-based system—despite Porter's attempts to outline just that.[4–7]

Several studies report fragmented and muddled efforts to implement VBHC.[8–10] Some scholars attribute these instances to the 'high level of abstraction' and 'vagueness' in which VBHC was originally described.[9] Although we recognise that VBHC is an abstract concept, we believe its muddled implementation can at least partially be explained by its multifaceted composition.

VBHC was developed as a strategy that aims to inform all stakeholders in healthcare systems.[4] It is an extensive concept with far-reaching implications: its goal is to 'transform health care' (p4).[4] In a value-based system, all stakeholders share a common objective: value for patients—with value defined as a patient's health status (outcomes) divided by the recourses it took to achieve that status (costs). Importantly, Porter and Teisberg argue that value can only be understood at the level at which it is created: in addressing a medical condition, over full cycles of care

(p5,99–105).[4] Providers should thus realign their organisational structures, forming integrated practice units which focus on one or a few related medical conditions and cover full care cycles (p167–77).[4] Payment structures should also be aligned with value, with bundled payments for full cycles (or episodes) of care (p265–67).[4] Perhaps most importantly (according to these scholars), providers should actively engage in benchmarking: they should systematically measure, report and compare their outcome data. This would fuel value-based competition, and enable patients, payers, providers and policy-makers to all make more value-based decisions.[4] In sum, VBHC encompasses numerous aspects and requires a whole range of actions and practices in order to be implemented.

In this paper, we aim to identify the relative importance of the various aspects of this multifaceted concept. This is both timely and important, because although the recent uptake of VBHC has been described as remarkable[3]; it nonetheless remains unclear what practical steps should be undertaken, and what aspects should be prioritised on the road towards a more value-based system. In fact, as mentioned earlier, several studies report muddled implementation efforts,[9 11] and it also appears that scholars employ different standards when they discuss the implementation of VBHC (cf. [12–14]). In addition, several scholars have stated that the way in which a multifaceted concept such as VBHC moves from idea to practice, is highly contingent on the particular intricacies within different health systems.[11 15] Thus, uncertainty prevails when it comes to the actual implementation of VBHC.

In this paper, we build on the Delphi method to identify the relative importance of various actions and practices in moving towards a value-based system in the Netherlands. The Dutch healthcare system is a particularly interesting case since it is based on regulated competition.[16] Moreover, the measurement and use of outcome data are increasingly becoming an important issue in Dutch healthcare policy.[16] Several of VBHC's aspects (as outlined by Porter) are thus already in place.

Accordingly, our research question is: which aspects, actions and practices do Dutch experts agree on as important in moving towards a value-based healthcare system?

## METHODS

The Delphi technique is a well-established research method to build consensus where considerable uncertainty exists, and where empirical evidence is (still) lacking.[17–20] In this modified Delphi study, we explore Dutch expert consensus on the most important aspects of VBHC, and the actions and practices that will contribute to implement VBHC in the Dutch system.

We recruited our expert panel through purposive sampling. Ten experts were selected based on their known or stated expertise regarding VBHC and the Dutch healthcare system. Nine panel members completed the first survey round: 4 females and 5 males who, at the time of the study, averaged nearly 23 years of experience in their current professional field, with 8 out of 9 members counting >10 years of experience regarding quality improvement. Additionally, these experts all have significant experience working with VBHC inspired initiatives, either through their profession within a hospital (n=5) or through their collaboration with healthcare organisations (n=4). Of the five participants working in a hospital, two are professors at an academic hospital, with a background in medicine; two are project leaders (VBHC); one is a manager (quality). Of those not directly employed by healthcare providers, one has a managerial function at a hospital association; the remaining three work in healthcare consultancy.

We created an initial list of 39 items (available on request). The bulk of these items were derived from the pioneering literature on VBHC.[4 6 21–24] We complemented this with several items that—particularly within Dutch health policy—are strongly related to VBHC. Accordingly, these items were extracted from policy documents that directly deal with one or more aspects of VBHC (eg, outcome measurements).[25–27] These complementary items are warranted, since our study builds on the notion that the implementation of VBHC will vary between health systems and sociopolitical regions.[11 15] Examples of item descriptions are: 'assessing the quality of a treatment cycle by measuring the achieved health status'; 'creating integrated practice units (IPUs)'; and 'learning from relating data on outcomes to data on costs of health care.'

Our expert panel completed questionnaires during a two-round modified Delphi survey, in which they rated each item according to 'how important you deem this item in moving towards a value-based healthcare system?' Scoring occurred on a four-point Likert scale: (1) 'very important', (2) 'important', (3) 'moderately important', (4) 'not important' (4). The first survey was sent out in December 2017, the second in January 2018. Panel members were given 3 weeks to complete each questionnaire.

In line with previous Delphi studies,[28] we retained items after each round that were rated as 'very important' (1), or 'important' (2), by at least 80% of the experts, and excluded those rated as 'not important' (4), or only 'moderately important' (3), by more than 50% the experts. We expect the distribution of scores to be skewed towards agreement on importance. Therefore, our threshold for agreement on importance (≥80% scores very important or important) is higher than for agreement on non-importance (>50%) scores moderately or not important.

Importantly, after rating an item, each expert was asked whether they had suggestions to reformulate that particular item. Additionally, by the end of the survey round, experts also had the possibility to add new items to the list, as they saw fit. Suggested additions and reformulations would become part of the next survey round. The second survey round, therefore, consisted of both the reformulated and unchanged items that scored between

Table 1 Results survey rounds 1 and 2

| Response | Round 1 (90 %) | Round 2 (80%) |
|---|---|---|
| Number of Items | 39 | 18 out of which: 5 unchanged 8 reformulated 5 new |
| *Consensus:* | | *Consensus:* |
| Included | 20 (45%) | 12 (66.7%) |
| Excluded | 6 (13.6%) | 0 (0.00%) |
| *Discordance:* | | *Discordance:* |
| Reformulated | 8 (18.2%) | 0 (0.00%) |
| Unchanged | 5 (11.4%) | 6 (33.3%) |
| Newly suggested items: | 5 (11.4%) | 0 (0%) |

inclusion and exclusion, plus the newly suggested ones from round one.

We thus conducted a *modified* Delphi study, particularly because we did not enable experts to revisit the aggregate scores of each item between survey rounds.[18] Since our goal was to generate consensus, we decided that only those items on which *no* consensus was reached in the first round would be presented to the panel again in the second round.

### Patient and public involvement statement

Within this study, there has been no involvement from patients or members of the public in the design, conduct, reporting or dissemination plans of the research.

### RESULTS

Table 1 shows the flow of our Delphi study. Of the 10 experts that were recruited, 9 (90%) agreed to participate and completed the study. Our analysis of the second round of questionnaires revealed missing data regarding one of the panel members; we therefore omitted this expert's data for the entire second round (80% response rate).

As the table shows, 20 items were included in the first round, that is, rated as important (2) or very important (1) by at least 80% of the panel members. Additionally, six items were rated 'moderately important' (3) or 'not important' (4) by more than 50% of the experts and were therefore excluded. This entails that *no* consensus was reached on 13 of our initial 39 items. These items thus became part of the second round, as did 5 new items put forth by panel members. In the second survey round, another 12 items were included by the panel members, bringing the total number of included items to 32 (20+12).

See table 2, for an overview of all 32 items that were included through expert consensus after 2 survey rounds. No consensus was reached on six items (see table 3 for an overview). However, in the second survey round experts did not suggest new items, nor did they suggest any reformulations—thus indicating saturation was reached.

Table 2 shows the 32 items that are included based on their consensually perceived importance in moving towards a value-based healthcare system. The items are rank ordered, first by mean ($\bar{x}$), second by SD (s). The mean ($\bar{x}$) indicates the average score of the item (ie, its perceived importance) according to our panel (rated by each member on a 4-point scale). An item's SD (s) was primarily used to rank order items with a similar mean, and can be regarded as a secondary indicator of overall agreement among panel members. The table also displays whether items were included in round 1 or 2.

According to experts, the most important practice in moving towards VBHC in the Netherlands is to involve patients in shared decision-making. Experts unanimously agree on the high importance of this item (#26). Other high ranking items on which experts agree are: to standardise performance measures for full treatment cycles of medical conditions (#21); to organise delivery of care around these full treatment cycles (#4); to use patient-reported outcome measures for evaluating care provision (#28); to use dashboards or scorecards to assess and visualise performance (#34); to learn how to optimise the relationship between health outcomes and costs (#43); and to assess the quality of care based on the patients' recovery process after treatment(s) (#23).

After two rounds of questionnaires, six items remained on which no consensus could be reached. In other words, these items were neither rated (very) important by ≥80% of the experts, nor were they rated moderately or not important by ≥50%. These six items are shown in table 3.

Experts did not reach consensus on the idea that the payment of healthcare delivery should be based on actual costs, rather than predetermined price rates (#40). Our panel also could not agree on the importance of the continual revision and improvement of standardised measures (#37), and the same applies to the repeated revision of general protocols and regulations (#18). Additionally, no consensus was reached on the importance of benchmarking based on outcome data (#39). Disagreement also remained regarding the issue of quality assessment based on the sustainability of a patient's health (#24). Similarly, experts did not agree on the importance of incentivising providers to improve their treatment outcomes (#31).

### DISCUSSION

Our Delphi study identified expert consensus on the relative importance of aspects, actions and practices in moving towards a value-based healthcare system. Consensus was reached on 32 items that are deemed important (table 2). In round 2, no new items were put forth, and there were also no suggestions for reformulation, thus indicating that saturation was reached. In the second round, six items remained on which experts did not agree sufficiently for either inclusion or exclusion.

**Table 2** Included Items (#1–#44) according to their mean importance score (x̄), SD (s) and round of inclusion (1 or 2)

| x̄ | s | Round | item | Item description |
|---|---|---|---|---|
| 1.00 | 0.00 | 1 | #26 | Involving patients in the shared decision-making process (regarding treatment options, etc) as much as possible. |
| 1.11 | 0.33 | 1 | #21 | Standardising performance measures for full treatment cycles of medical conditions, rather than for individual treatments/procedures. |
| 1.22 | 0.67 | 1 | #4 | Organising delivery of care around full treatment cycles of medical conditions, rather than around individual procedures. |
| 1.33 | 0.50 | 1 | #28 | Using patient-reported outcome measures to evaluate the provision of care. |
| 1.33 | 0.50 | 1 | #34 | Using dashboards or scorecards to assess and visualise performance. |
| 1.38 | 0.52 | 2 | #43 | Learning to optimise the relationship between health outcomes and costs. |
| 1.38 | 0.52 | 2 | #23 | Assessing the quality of the provided care based on the patients' recovery process after treatment(s). |
| 1.44 | 0.73 | 1 | #12 | Delivering a desired and sustainable outcome from a patient's perspective, rather than an optimal outcome from a practitioner's perspective. |
| 1.44 | 0.73 | 1 | #9 | Including a patient representative in the improvement team in order to ensure expert input from the patient perspective. |
| 1.44 | 0.73 | 1 | #20 | Reducing the amount of performance measures that are used. |
| 1.44 | 0.73 | 1 | #35 | Learning from relating data on outcomes to data on costs of health care. |
| 1.50 | 0.53 | 2 | #5 | Developing a technological/digital platform that can be used to view data and share data with others, with the aim of improving the provision of care. |
| 1.56 | 0.53 | 1 | #27 | Establishing clear and realistic expectations for patients. |
| 1.56 | 0.53 | 1 | #16 | Reducing waste (eg, the waste of time, material and/or staff capacity). |
| 1.56 | 0.73 | 1 | #13 | Ensuring the general safety of patients when undergoing treatment. |
| 1.63 | 0.52 | 2 | #2 | Striving to make individual health insurance as affordable as possible. |
| 1.63 | 0.74 | 2 | #41 | Describing the care process in care pathways, in which the goals and the 'evidence-based' key interventions (who does what, and at what time) are established. |
| 1.63 | 1.06 | 2 | #1 | Providing or aiming to provide universal coverage (health insurance). |
| 1.67 | 0.71 | 1 | #17 | Creating integrated practice units |
| 1.67 | 0.71 | 1 | #6 | Assigning a data or business intelligence manager (or team) who focusses on collecting and analysing existing data from patient records. |
| 1.67 | 0.71 | 1 | #14 | Avoiding over and underuse of healthcare services. |
| 1.67 | 1,00 | 1 | #22 | Assessing the quality of a treatment cycle by measuring the achieved health status. |
| 1.67 | 1,00 | 1 | #30 | Structuring payments for health care so that they cover the costs of a full cycle of care, rather than having separate payments for individual procedures. |
| 1.75 | 0.71 | 2 | #7 | Developing a standardised step-by-step plan (roadmap) that healthcare providers can use to transition into value-based providers. |
| 1.75 | 0.71 | 2 | #8 | Appointing a change manager (an expert in the field of value-based health care) who helps healthcare providers transition into 'value-based' providers. |
| 1.75 | 1.04 | 2 | #29 | Using patient-reported experience measures to evaluate the provision of care. |
| 1.78 | 0.67 | 1 | #10 | Using a patient's physical well-being in assessing the outcome of healthcare delivery. |
| 1.78 | 0.67 | 1 | #38 | Creating predictive models that enable medical specialists to provide information concerning a patient's future health status. |

Continued

**Table 2** Continued

| x̄ | s | Round | item | Item description |
|---|---|---|---|---|
| 2.00 | 0.50 | 1 | #25 | Choosing and adapting indicators from ICHOM sets (standardised outcome measurements for various medical conditions). |
| 2.00 | 0.53 | 2 | #44 | Identifying and removing the barriers raised by privacy legislation that obstruct the path towards value-based healthcare delivery. |
| 2.00 | 0.93 | 2 | #11 | Using the patient's mental well-being as an outcome indicator in assessing healthcare services. |
| 2.00 | 0.93 | 2 | #42 | Striving to standardise outcome indicators in such a way that different groups of patients can be compared with one another. |

Our most eye-catching finding concerns the consensus on the importance of shared decision-making (SDM). Experts unanimously rated this particular item (#26) as 'very important' in moving towards a value-based system—which demonstrates a unique level of agreement, unmatched by any other item in this study. Interestingly, SDM is by no means a fundamental aspect within the pioneering literature on VBHC.[4–6 22] In contrasts to SDM, which specifically concerns the deliberate discussion of treatment options, this body of work emphasises the value-adding options patients have (or should have) in choosing among healthcare providers. Recently, it has been argued that the original VBHC concept, and the framework of market-based choices on which it rests, deemphasises patients' personal values in life.[3] Perhaps our panel's unanimous agreement indicates that the incorporation of SDM may add a more personal dimension to VBHC—which has been advocated by some scholars.[29]

In addition, multiple items reveal that experts agree on the importance of recognising *full care cycles for medical conditions* as the relevant level of analysis in health care. This applies to the organisation of healthcare delivery (#4 and 17), its performance measurements (#21), and its payment structures (#30). This resonates with the literature on VBHC, particularly with the work of Porter, who repeatedly states that value in health care is created at the level of medical conditions, over full cycles of care.[4 6 30]

Several items on which consensus was reached relate to the importance of outcome information (eg, #22, 25, 28). Our panel agreed, for instance, that it is important to assess the quality of a treatment cycle by measuring the achieved health status (ie, outcomes) of patients (#22). This overall emphasis on outcome measurement also corresponds with the literature.[4 22 24]

Regarding outcomes, this correspondence may seem relatively straightforward, since the central tenet of VBHC is that all stakeholders must aim to improve value for patients, with value defined as health outcomes per unit of costs.[4 7] However, our panel did *not* display similar correspondence regarding costs—the denominator of value ($value = \frac{outcomes}{costs}$). Dutch experts thus appear to prioritise measuring outcomes over measuring costs, which may reflect other studies that indicate that when VBHC is being implemented, the costs of care attain relatively little attention.[8 31]

One of the items on which our panel did not agree concerns the importance of comparing and benchmarking provider's performance data (#39). Accordingly, and strikingly, experts did not reach consensus regarding the importance of one of the most foundational aspects of VBHC theory:

> Providers need to be compared on results, and excellent providers rewarded with more patients. Information about results [outcomes versus costs], which is appropriately risk adjusted, must become the critical driver of behaviour in the system—by

**Table 3** Items with expert discordance after two survey rounds, according to their mean importance score (x) and SD (s)

| x̄ | s | Item | Item description |
|------|------|------|------------------|
| 1.63 | 1.19 | #31 | Applying an incentive structure that stimulates providers to improve outcomes of care, rather than increasing volume. |
| 1.75 | 0.89 | #18 | Updating and reformulating protocols and regulations iteratively in order to improve the quality of care. |
| 1.88 | 0.83 | #24 | Assessing the quality of a treatment cycle based on the sustainability of a patient's health. |
| 1.88 | 0.83 | #39 | Comparing the data of different IPUs or multidisciplinary teams in order to benchmark performance. |
| 2.00 | 0.76 | #37 | Revising and reformulating existing measures continuously, and continuously developing new measures that are used to evaluate healthcare delivery. |
| 2.38 | 0.92 | #40 | Basing the payment of healthcare services on the actual costs, and not on prearranged rates. |

referring physicians, by health plans, by patients, and by providers themselves.[4] (p102)

Faced with the challenge to establish a value-based system in the Netherlands, it appears that although Dutch experts agree on the importance of multiple aspects of Porter's original conceptualisation of VBHC, they also blend in additional concepts (eg, SDM), while bypassing others (eg, benchmarking). It will require additional research, however, to determine the extent to which our study represents the range of Dutch expert opinion on this issue.

### Limitations

One potential limitation of this study is that our panel consisted entirely of *Dutch* experts. However, we were interested in the implementation of VBHC in the Dutch system, and it therefore made sense to invite Dutch experts to participate. Accordingly, this has enabled us to demonstrate how, in the Netherlands, VBHC is being adapted and blended with other concepts such as shared decision-making. Additionally, experts might have been influenced by the particular items that were first presented to them, and this could have affected their scoring. To counterbalance this potential bias, however, experts could reformulate existing items, while also being able to suggest new ones as they saw fit—both of which they did (see table 1).

### CONCLUSION

In this paper, we identified expert consensus on the relative importance of a variety of concepts and practices for moving towards a value-based healthcare system. Accordingly, our study provides additional insight regarding several important steps within the implementation of VBHC—a topical concern within many healthcare systems. However, our study also reveals considerable contrast with the pioneering literature on VBHC. Perhaps our results, based on a Dutch expert panel, are a precursor to a process of implementation of VBHC in the Netherlands that deviates from the original concept—which has been observed elsewhere.[8 9] In such circumstances, some scholars have questioned whether VBHC is actually being implemented or, on closer look, primarily serves as an inspiring idea.[31]

**Contributors** GS: data analysis and interpretation, drafting and completing the manuscript. DD: drafting and completing the manuscript. HvdB: assisted in drafting and completing the manuscript. RG: assisted in study design, data collection. KA: study design, assisted in data collection, assisted in interpretation of data, assisted in drafting and completing the manuscript. All authors read and approved the final manuscript.

**Funding** Zorginstituut Nederland (National Health Care Institute) funds this research (reference number 2018007737). There has not been any involvement of this funding body in the design of the study; collection, analysis, and interpretation of data; nor in writing the manuscript.

**Competing interests** The fourth author, DD, is professor at the Erasmus University Rotterdam while also employed at Zorginstituut Nederland (National Health Care Institute), which partially funds this research.

**Patient and public involvement** Patients and/or the public were not involved in the design, or conduct, or reporting, or dissemination plans of this research.

**Patient consent for publication** Not required.

**Ethics approval** Dutch legislation on Medical Research Involving Human Subject (WMO) does not require ethical approval for this project, as the research would not contribute to clinical medical knowledge and no participation by patients or use of patients' data was involved. All participants gave informed consent before taking part in the study.

**Provenance and peer review** Not commissioned; externally peer reviewed.

**Data availability statement** Data are available upon reasonable request. The data set that was analysed for this study is available from the corresponding author upon reasonable request.

**ORCID iD**
Gijs Steinmann http://orcid.org/0000-0002-9478-3725

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
