## [Reviewer comments · BMJ Open]

ARTICLE DETAILS

TITLE (PROVISIONAL)	Expert consensus on moving towards a value-based healthcare system in the Netherlands: a Delphi study
AUTHORS	Steinmann, Gijs; Delnoij, Diana; van de Bovenkamp, Hester; Groote, Rogier; Ahaus, Kees

VERSION 1 – REVIEW

REVIEWER	Richard Lewanczuk Alberta Health Services Canada
REVIEW RETURNED	06-Oct-2020

GENERAL COMMENTS	This is a well written and straightforward paper. It is of great interest to those of us who work in the area of value-based healthcare at a large systems area - we are interested in international variance. I have some specific comments below but am providing a more general comment which the authors may want to take into account in any revision. Although I think this is an important point to highlight or discuss, it is actually not a major revision issue as it represents my opinion. The paper could stand with just the minor revisions suggested below. My personal perspective is that the authors come close to, but just miss the point of VBHC being contextual. That is, it means different things to different populations and health systems. In other words, residents of the Netherlands may value certain aspects of healthcare differently than Swedes or Canadians. This is quite fine and is the beauty of VBHC. In this paper, it tends to come across as if Porter set unalterable, permanent "rules" for VBHC, and if a health system doesn't adhere to these rules they don't achieve VBHC. In the system where I work, it is not all about clinical outcomes/cost. Rather, it is the total patient experience/cost which is important, bringing in some aspects which came out in the Delphi: patient experience (via PROMs and PREMs), mental health outcomes, patient perspective on outcome rather than provider, etc. Thus, the VBHC as defined by the Dutch experts in this paper is closer to the definition my system would choose versus the criteria of Porter. I think the freedom for a system to define VBHC in the context of what is important to its population is important to both highlight and celebrate. Minor comments are as follows:
--

	1. In line 93, although cited by Porter, the comment on payment models seems out of context and confusing, thus I would eliminate it. 2. A bit more detail on the expert panel should be included. Were these experts familiar with the concept of VBHC? How many were leaders/administrators versus clinicians - or were they clinician leaders? What was their background? This information will be helpful in setting the context of the results. 3. The description in lines 177-179 is confusing and I really couldn't follow. It should be clarified. 4. In Table 2 or the narrative results, I am not sure as to the purpose of including item numbers. Is there some significance to these or can they be eliminated? 5. In Table 2 or perhaps more appropriately the mean importance score should be explained. Similarly, there should be an explanation of the standard deviation. Is it based on all 4 potential ratings? I am not sure as to the validity given just a 4 point scale. I presume this is used as an indicator of unanimity of opinion. This may be obvious to many, but it bears mentioning, if this is the case. Similarly, if they include the data, then the authors may want to comment on items which had high unanimity but high vs low SDs. (personally with just a 4 point scale I'm not sure it adds much)
--	--

REVIEWER	Chester Good University of Pittsburgh Division of General Internal Medicine Pittsburgh, PA USA
REVIEW RETURNED	25-Oct-2020

GENERAL COMMENTS	This manuscript provides useful information by confirming important considerations for the successful implementation of a value-based healthcare system, using items extracted from the literature, and adding to that group of recommendations. Specifically, the most important item found (shared decision-making) was a particularly useful finding- one that was not universally accepted in previous reports. The paper refers to this study as using Delphi methodology. While I am not an expert on Delphi methodology, it seems that this study is more consistent with an iterative simple survey, rather than a Delphi survey. As I understand Delphi methodology, important items are identified (either through review of the literature, or by surveying experts), and then experts rank the importance of those items. Subsequent rounds of the Delphi survey present the findings of the initial round, and allow the experts to re-rank the items in light of how other experts have responded and to consider whether to change the ranking of their responses. As I understand this study, round 1 consisted of 39 items of which 20 reached consensus (as defined by researchers), 6 items were not rated as sufficiently important to be considered in the next round, 8 items were "reformulated" by at least one respondent, and 5 new items were identified by experts. Importantly, for round 2, only the reformulated questions, new questions, and questions where there was discordance among experts were included for
--

	ranking. Thus, experts were not given the opportunity to consider the results of those 20 questions that were not included in round 2. I reviewed the articles referenced for Delphi studies, which reflect my understanding of Delphi methodology. Ref 20 (Jones) describes the the Delphi process as where participants (experts) re-rank their agreement with each statement/item based on a "repeat version of the questionnaire". Ref 18 (Hasson) describes the process as being an "iterative multi-stage process designed to transform opinion into group consensus." Regardless of my concerns about whether this study truly followed Delphi methodology, the findings are useful and interesting. I defer to the editors and experts on Delphi methodology to decide if my concerns are valid and require any discussion in the Discussion/Limitations. The use of the term "included" throughout the manuscript was confusing to me. In the manuscript "included" means those questions that reached adequate consensus as defined by the authors- but as I read the manuscript I kept associating the term with which questions were included in the survey and not necessarily reaching consensus. For example, lines 177-178 discuss 12 items "included" in the second round- but I confused this with all 18 items that were reviewed by experts. Perhaps this reflects a difference in use of this term among readers from different countries, but I think that U.S. readers will find this use of "included" to be confusing. For me, "reached consensus" would be less confusing. On a related note, I found Table 1 to be confusing. It took me a while to get the numbers to add up, probably reflecting my confusion with terminology. If it could be reformatted, it would help this reader. This study allowed experts to reformulate questions, which were then included in the 2nd round. There are no details of how many experts ranked the original question (un-reformulated) - and whether that original question had high levels of agreement from experts. Thank you for allowing me to review this interesting paper.
--	---

VERSION 1 – AUTHOR RESPONSE

Reviewer 1. Before addressing the general comment made by the reviewer more fully, we will first respond to the specific comments aimed at minor revisions.

1. "In line 93, although cited by Porter, the comment on payment models seems out of context." – Although we have considered and discussed removing the line on payment models, we believed this to be relatively important element within Porter's VBHC-theory, and this topic is also incorporated within the original items of our Delphi survey. Therefore, we have revised the sentence (page 4, line 94-5), indicating that Porter envisions payment structures that are aligned with value creation: bundled payments that cover the full cycle of care for a particular medical condition. We hope our revisions will properly place the adapted line on payment structures in the context of Porter's VBHC-theory.
2. "A bit more detail on the expert panel should be included." We have added supplementary background information (page 6, line 133-7).
3. "The description in lines 177-179 is confusing and [...] should be clarified." We realize that our description needed clarification. We believe we have revised the respective sentence, now highlighted on page 9, line 185-7.

4. The reviewer inquired about the utility of displaying item numbers in tables 2 and 3. In the main text, we use these item numbers to refer to the particular items in abbreviated fashion. Additionally, we believe this would make it easier for readers to go back and forth between text and tables.

5. Regarding table 2 and in extension table 3, the reviewer stated “the mean importance score should be explained. Similarly, there should be an explanation of the standard deviation.” We have revised the explanation below table 2, with the additional explanation highlighted on page 11, line 198-201. With regard our use of the standard deviation (SD), we primarily used this as a tool to rank order those items which had a similar mean score.

We fully agree with the reviewer’s statistical remarks: using SD as an indicator of overall agreement can be problematic, particularly considering the 4-point scale. With such a controlled range of options the mean is much more informative. We do believe, however, that within this study, aimed at generating consensus, SD might serve as a secondary/accessory indication of consensus.

Nevertheless, as mentioned, SD was primarily used as a complementary tool to rank order items. We hope our revisions in lines 198-201 will demonstrate this.

Next to the minor revisions discussed above, reviewer 1 also had a more general comment. To summarize, it was stated that “the authors come close to, but just miss the point of VBHC being contextual. That is, it means different things to different populations and health systems. [R]esidents of the Netherlands may value certain aspects of healthcare differently than Swedes or Canadians. [I]n this paper, it tends to come across as if Porter set unalterable, permanent “rules” for VBHC, and if a health system doesn’t adhere to these rules they don’t achieve VBHC. [I] think the freedom for a system to define VBHC in the context of what is important to its population is important to both highlight and celebrate.”

We agree that the freedom for a system or population to interpret and adapt/translate VBHC according to its particular socio-political context is indeed something to be cherished. We recognize and share this point of view, and sincerely appreciate the reviewer expressing this so clearly. Regarding this particular paper, however, highlighting this viewpoint has not been our focus – nor do we intend to advocate in favour of or against any kind or degree of adaptation. Our focus was on generating consensus regarding the importance of several aspects and actions toward a vbhc-system the Netherlands. Both within our methodology (e.g. formulating the 32 original items) and within our analysis we used Porter’s theory as a reference point, rather than “unalterable, permanent rules.” In doing so, we partially framed our analysis through a comparison between Dutch expert consensus on the one hand and Porter’s theory on the other. This, we believe, led to some interesting findings (regarding benchmarking and shared decision-making in particular).

We fully agree that VBHC (similar to most, if not all concepts) means different things to different people. And although this viewpoint may not have been elaborated on extensively within this particular paper, we do believe that the paper recognizes this at several instances:

- Page 5, line 107-9 (“the way in which a multifaceted concept such as VBHC moves from idea to practice, is highly contingent on the particular intricacies within different health systems”).

- Page 6-7, line 142-4 (“our study builds on the notion that the implementation of VBHC will vary between health systems and socio-political regions”).

- And in the newly added sentence on page 3, line 62-3 (“this enabled participants to express their personal (re)interpretation of value-based health care”).

In sum, we share the reviewer’s perspective, and hope that our response here has sufficiently addressed the respective issue.

Reviewer 2.

1. The reviewer questioned our use of the label “Delphi”, particularly since, strictly speaking, the Delphi technique “present[s] the findings of the initial round, and allow the experts to re-rank the items in light of how other experts have responded and to consider whether to change the ranking of their responses.”

Indeed, as the reviewer noted, within our study the panel members were not offered the opportunity to revisit the scores of those items on which they reached consensus in the first survey round, before starting the second. For this reason, we have previously considered using the label “modified Delphi study.” We eventually chose not to do so, since the definition of the method appears to have somewhat broadened/loosed, with other studies similar to ours using the “Delphi” label, without mentioning modifications to the original definition (e.g. Minkman et al. 2009 (reference 18): <https://doi.org/10.1093/intqhc/mzn048>). However, we agree with the reviewer’s comments on this issue, and we have therefore labelled our study a “modified Delphi”:

- Page 6, line 123

- Page 7, line 147

- And an explanation for this modification on page 7-8, line 165-8.

2. Regarding our use of the term “included”: “the use of the term “included” throughout the manuscript was confusing. In the manuscript “included” means those questions that reached adequate consensus as defined by the authors- but as I read the manuscript I kept associating the term with which questions were included in the survey and not necessarily reaching consensus.”

Although we understand and recognize this point, we believe our use of the terms “included” and “excluded” are in accordance and consistent with most Delphi studies (at least those familiar to us). Using “reached consensus” as an alternative would pose problems, as we also speak of consensus when items are agreed upon as not important – which were “excluded.” In order to prevent any potential confusion, we have screened the manuscript: whenever we use “included” it refers to one or more items being consensually perceived as (very) important by our panel.

3. “On a related note, I found Table 1 to be confusing. It took me a while to get the numbers to add up, probably reflecting my confusion with terminology. If it could be reformatted, it would help this reader.” We have revised table 1 (page 7-8, line 179-80) – the additions/changes are highlighted (yellow). We hope this helps clarify the content.

4. “This study allowed experts to reformulate questions, which were then included in the 2nd round. There are no details of how many experts ranked the original question (un-reformulated) - and whether that original question had high levels of agreement from experts.” All 39 original items from round 1 were rated by all 9 panel members (see table 1 on page 8). Since our aim was to generate consensus, we only considered recommended reformulations for items on which no consensus was reached. So, the 8 items from round 1 that became part of round 2 – in reformulated fashion – did not reach consensus in round 1 (see table 1 page 8, and line 162-4 on page 7). We hope this clarifies this final remark.

We would like to express our sincere gratitude to the sharp comments and critical questions posed by the reviewers and the editor; this enabled us to improve the manuscript accordingly. We hope you will find your concerns addressed adequately, and we look forward to your response.

VERSION 2 – REVIEW

REVIEWER	Richard Lewanczuk University of Alberta and Alberta Health Services Canada
REVIEW RETURNED	22-Feb-2021

GENERAL COMMENTS	My previous concerns and issues have been addressed in this revision. I have no further suggestions.
--

REVIEWER	Chester Good University of Pittsburgh School of Medicine
-----------------	---

	USA
REVIEW RETURNED	17-Feb-2021

GENERAL COMMENTS	I have previously reviewed this paper (see previous comments). I have read the authors comments to my suggestions, as well as other reviewer. I have also re-read the manuscript and reviewed the changes. I believe that the authors have responded to those suggestions, and have modified the manuscript accordingly. At this point I have no further suggestions, and recommend accepting the manuscript.
--